# Sleep-Related Breathing Disorders in Children—Red Flags in Pediatric Care

**DOI:** 10.3390/jcm11195570

**Published:** 2022-09-22

**Authors:** Sigalit Blumer, Ilana Eli, Shani Kaminsky-Kurtz, Yarden Shreiber-Fridman, Eran Dolev, Alona Emodi-Perlman

**Affiliations:** 1Department of Pediatric Dentistry, The Maurice and Gabriela Goldschleger School of Dental Medicine, Sackler Faculty of Medicine, Tel Aviv University, Tel Aviv 6139001, Israel; 2The Maurice and Gabriela Goldschleger School of Dental Medicine, Sackler Faculty of Medicine, Tel Aviv University, Tel Aviv 6139001, Israel; 3Department of Oral Rehabilitation, The Maurice and Gabriela Goldschleger School of Dental Medicine, Sackler Faculty of Medicine, Tel Aviv University, Tel Aviv 6139001, Israel

**Keywords:** sleep-related breathing disorders (SRBD), Pediatric Sleep Questionnaire (PSQ), snoring, ADHD, mouth breathing

## Abstract

Objectives: In recent years, we have witnessed a growing interest in pediatric sleep-related breathing disorders (SRBD). Although a Pediatric Sleep Questionnaire (PSQ) exists and was found reliable in screening SRBD in children, many of the children remain underdiagnosed. The aim of the present study was to define anamnestic and clinical findings that can serve as red flags indicating the presence of SRBD in children. Methods: 227 children aged 4–12 years old were evaluated with regard to the following parameters: (i) anamnestic variables (e.g., general state of health, oral habits, bruxism, esophageal reflux, sleep continuity, snoring); (ii) clinical parameters (e.g., oral mucosa, palate, tonsils, tongue, floor of the mouth, angle classification, gingival health, caries risk) and (iii) presence of SRBD (through the PSQ). Results: Significant differences between children with and without SRBD were observed regarding continuous sleep, developmental delay, mouth breathing, and snoring. Taking medications for ADHD increased the odds of SRBD in children by over seven times, non-continuous sleep increased the odds of SRBD by six times, mouth breathing increased the odds by almost five times, and snoring increased the odds by over three times. Conclusions: Child caregivers from various fields (dentists, orthodontists, pediatric physicians, school nurses) should actively inquire about disturbed sleep, medications for ADHD, snoring, and mouth breathing among their young patients. Initial screening through a few simple questions may help raise red flags that can assist in the early detection of SRBD in children and lead to proper diagnosis and treatment.

## 1. Introduction

Sleep medicine has emerged as an important field in pediatric care. Its main focus is on sleep-related breathing disorders (SRBD). SRBD is a vast term that includes conditions ranging from frequent snoring to obstructive sleep apnea (OSA). On one side of the SRBD spectrum, with high prevalence, lies the familiar symptom of snoring [1]. The other side of the SRBD spectrum belongs to OSA, which is characterized by a repetitive collapse of the upper airway during sleep, often associated with oxygen desaturation and/or arousal from sleep [2]. 

The criteria for diagnosis of childhood OSA are either one or more obstructive events per hour of sleep or obstructive hypoventilation, together with snoring, paradoxical thoraco-abdominal movement, or flattening of the nasal airway pressure waveform implying flow limitation [3]. Clinical categories are commonly defined through an obstructive apnea-hypopnea index [4]. 

Although it is not possible to accurately estimate the prevalence of childhood OSA, a minimum prevalence of 2% to 3% is likely, with prevalence as high as 10% to 20% in habitually snoring children [5]. Studies estimate that approximately 8–27% of children suffer from frequent snoring, and 1–5% of children suffer from OSA [6,7]. The diagnosis is often associated with snoring, with a round of 70% of the children diagnosed with SRBD also diagnosed with primary snoring [8]. 

SRBD can be detrimental to a child’s health and quality of life. It can cause excessive daytime sleepiness, behavioral problems, learning disabilities, right-sided heart failure, and even growth retardation [9,10,11]. Children with habitual snoring showed a higher prevalence of recurrent otitis media [12], while children with otitis media with effusion experienced significant symptoms of OSA and associated impairment of quality of life [13]. Isaiah et al. [14] showed an association between regional structural alterations in cortical gray matter and problem behaviors reported in children with obstructive sleep-disordered breathing. Associations have been found between SRBD in children and several anatomical features such as enlarged tongue, tonsils or uvula, high soft palate, narrow dental arches, teeth crowding, and upper airway blockage [15,16].

Literature review showed that dentists play a significant role in the early detection of OSA in children, helping in reducing and preventing its serious consequences [17]. The authors point out that a multidisciplinary treatment team, including a dentist in addition to a sleep specialist and an ENT physician, should carry out managing pediatric OSA. In 2017, the American Dental Association (ADA) defined the role of dentistry in the treatment of SRBD and encouraged dentists to screen patients for the syndrome as an integral part of routine dental care. The ADA recognized the growing prevalence of SRBD in pediatric dental patients and pointed out the importance of screening through history and clinical examination [18].

The gold standard in SRBD diagnosis remains polysomnography. However, the high expense and intricacy involved with taking a young child to a sleep laboratory result in low parental compliance.

Over recent years, several sleep questionnaires have been developed to screen for SRBD [19,20]. One of the questionnaires, specially developed for children, is the Pediatric Sleep Questionnaire (PSQ). The PSQ was found to be a valid and reliable instrument that can be used to identify SRBD or associated symptom constructs in clinical research when polysomnography is not feasible [21]. It has been used in numerous studies worldwide [22,23,24,25]. A recent meta-analysis comparing different SRBD screening questionnaires found that PSQ performed well and was the most sensitive screening questionnaire using the diagnostic threshold of apnea-hypopnea index (AHI) ≥ 1 for pediatric OSA [26].

In spite of the growing interest in pediatric SRBD, many of the children who have SRBD remain undiagnosed [25,27]. The most common reason for under-diagnosis lies in the fact that children frequently present signs and symptoms that are less widely recognized, such as neurobehavioral, cardiovascular complications, and growth impairment [28,29].

The aim of the present study was to define anamnestic and clinical findings that can serve as red flags indicating the presence of SRBD in children.

## 2. Materials and Methods

### 2.1. Population

Children arriving for treatment at the Department of Pediatric Dentistry, School of Dental Medicine, Tel Aviv University, in the years 2020–2022 were evaluated (cross-sectional design).

Parents (or legal guardians) of children aged 4–12 years were requested to grant their informed consent to participate in the study. The age of 4 years was selected as the youngest age for inclusion because it is the habitual age for the diagnosis and treatment of enlarged tonsil problems [30]. The age of 12 was selected as the oldest age for inclusion because it is considered the average age for starting orthodontic treatments.

Exclusion criteria were children with abnormalities and/or systemic diseases that might affect the study (e.g., autistic spectrum disorders, mental retardation, cerebral palsy, etc.), children under orthodontic treatment at present or in the past.

The study was approved by the Ethics committee of Tel Aviv University (no. 0001104-1).

### 2.2. Data Collection

#### 2.2.1. Anamnesis

An anamnestic questionnaire was distributed to parents prior to the child’s initial examination. The questionnaire included information regarding the child’s state of health (diagnosed medical conditions, regular medications, known allergies, previous hospitalization(s) or surgeries, etc.), information about the course of pregnancy and delivery, oral habits, sleep bruxism (is your child grinding his/her teeth during sleep), presence of esophageal reflux, child’s sleep continuity, snoring.

Additionally, parents were asked to report their own possible sleep bruxism, snoring, or being diagnosed with SRBD (does any one of the parents grinds his/her teeth during sleep or snores, was diagnosed with sleep-disordered breathing, or is suffering from breathing allergies or obstructed nasal airway).

#### 2.2.2. Pediatric Sleep Questionnaire (PSQ)

A Hebrew validated version of the PSQ [31] was included as an integral part of the forms given to the accompanying parent. In order to decrease the possibility of bias, upon completion and prior to the clinical examination, the PSQ was separated from the rest of the forms by the secretarial staff.

The PSQ contains 22 yes/no questions referring to three symptom categories: snoring and breathing problems, daytime sleepiness, and behavioral symptoms. The PSQ shows high sensitivity and specificity when 8 or more questions are answered as positive. This test criterion identifies children with excessive negative intrathoracic pressures and children with obstructive sleep apnea [21].

#### 2.2.3. Clinical Examination

Clinical examinations were performed by dentists specializing in pediatric dentistry and supervised by certified specialists in the field. Being part of an official internship program, all dentists were strictly calibrated in the examining procedure. The following clinical data were collected: examination of the oral mucosa, palate, tonsils, tongue (including size, location, tongue tie, etc.), floor of the mouth, angle classification for deciduous and permanent dentition, gingival health, caries, and additional dental diagnoses such as hypo-mineralization, agenesis of teeth, and others.

### 2.3. Statistical Analyses

Data were analyzed using SPSS software (IBM SPSS statistics 27.0, Armonk, NY, USA). Chi-square and t-test analyses were used to compare the groups with regard to collected data (age, BMI, general health status, oral behaviors and habits, parental self-report). In order to avoid the probability of type I error, Bonferroni correction was applied. Bonferroni adjusted alpha levels for variables referring to the child self were set at 0.001. Bonferroni adjusted alpha levels for variables referring to the parents were set at 0.01.

In a second step, a forward stepwise logistic regression was used to determine which of the variables could predict the presence of SRBD in children (SRBD positive). Potential multicollinearity in the logistic regression model has been evaluated using Fisher’s exact test and Phi coefficient.

## 3. Results

A total of 302 parents or children’s legal guardians provided their informed consent for the participation of their children in the study. A total of 75 children were excluded due to medical conditions or incomplete PSQ. The final study group consisted of 227 children.

As suggested by Chervin et al. [21], a cutoff point of ≥8 was used to define children with SRBD positive (SRBD positive, PSQ ≥ 8) versus children with no SRBD (non-SRBD, PSQ < 8).

No differences in age, sex, BMI, or factors referring to birth or to the number of children in the family were found between the SRBD positive and the non-SRBD groups. As expected, there was a significant difference between groups in their mean PSQ score (Table 1).

### 3.1. Descriptive Statistics

The general health status of the two groups is presented in Table 2.

Following Bonferroni correction, significant differences (*p* < 0.001) between groups were found only in the variables continuous sleep and developmental delay.

Children’s oral behavior and habits are presented in Table 3.

Following Bonferroni correction, significant differences (*p* < 0.001) between groups were found in the variables mouth breathing and snoring.

There were no differences between groups with regard to intra-oral oral findings such as anatomy of the palate, tonsils, tongue or floor of the mouth, angle classification for deciduous and permanent dentition, caries risk assessment, gingival health, hypo-mineralization, or other dental diagnoses.

The pregnancy of about 96% of the children in both groups was uneventful. There were no complications in the delivery process for 92.6% of the children with no SRBD and 95.8% of the children with positive SRBD. No differences between groups were found with regard to the parameters of child pregnancy and delivery.

Parental self-report regarding their own sleep bruxism, snoring, or being diagnosed with SRBD is presented in Table 4.

Following Bonferroni correction, a significant difference (*p* < 0.01) between groups was found only in the variable of parental snoring.

### 3.2. Multivariate Analysis

In an effort to determine which of the variables increases the odds of SRBD in children, a forward stepwise logistic regression was used. Variables entered into the equation were the child’s variables of general health, oral behavior, and habits. Variables referring to parental self-report were not included in the calculation.

Results show that taking medications for ADHD, non-continuous sleep, mouth breathing, and snoring significantly increased the odds of SRBD in children (Table 5).

Fisher’s exact test revealed positive associations between taking medications for attention deficit hyperactivity disorder (ADHD) and mouth breathing (*p* < 0.01) and between mouth breathing and snoring (*p* < 0.001). The Phi coefficient, which is less sensitive to sample size, revealed associations between taking medications for ADHD and mouth breathing (Phi = 0.216, *p* < 0.005) and between mouth breathing and snoring (Phi = 0.386, *p* < 0.001).

In spite of the associations, each of the variables showed a significant contribution to the model. The accuracy of the model was 90.3%.

## 4. Discussion

SRBD is a common finding in the pediatric population. Its highest incidence is between the age of 2–8, probably due to the relative size of lymphoid tissue in comparison with airway diameter [32,33,34,35].

About 10% of the children in the present study were SRBD positive, as defined by the PSQ. This coincides with other studies in which PSQ was used to study SRBD in children [22]. This high prevalence should be in the mind of any pediatric caretaker. The impact of SRBD on oral health and health-related quality of life in children is relevant and far-reaching [36].

When a diagnosis of SRBD arises, some of the parents wonder if there might have been some disturbance during pregnancy or delivery that led to the development of the syndrome. The literature on this topic is controversial [34,37,38]. Present results found no confirmation of such a notion. The finding that parents of children with SRBD suffer more from snoring than parents of the non-SRBD group is intriguing. Sleep bruxism and snoring have been associated with SRBD in adults [39,40]. Bruxism, as well as an association between sleep bruxism and sleep apnea, may be (at least in part) genetically determined [41,42]. Segu et al. reported a significant correlation between parental-reported tooth grinding and several sleep disorders concerning bedtime problems, night awakenings, nocturnal symptoms, and morning symptoms among their children [43]. Further studies are necessary to define the possibility of familial characteristics that link child SRBD to parental SRBD.

The present findings show that children with SRBD differ from children without SRBD with regard to some general health factors such as hearing problems and enlarged tonsils. Although not statistically significant, these findings are in accord with previous publications and are probably originating in a shared etiology of adeno-tonsillar hypertrophy [44,45]. DaRocha et al. showed that dental clinical parameters, including bruxism, mouth breathing, and history of tonsillectomy, were associated with higher PSQ scores [25]. In the present study, children with SRDB showed less continuous sleep and more developmental delay than their non-SRDB counterparts.

The finding that around one-quarter (25%) of the children with SRBD are taking medications on a regular basis and 30% of them are suffering from either developmental delay or behavioral disorders is alarming. The fact that many of these children may be underdiagnosed [27] can cause long-term deleterious health problems that may accompany the children for many years.

An important aspect of underdiagnosing pediatric SRBD is the possibility of the prescription and use of medications that are not necessarily adequate for the child. Chervin et al. [46] suggested that SRBD, and perhaps other sleep disorders, could be a cause of inattention and hyperactivity in some children. Since then, several studies have confirmed the presence of cognitive and behavioral consequences (e.g., deficits in neurocognitive performance, behavioral impairments, decreased school performance) in subjects with SRBD [47,48].

ADHD is a common neurodevelopmental disorder in childhood. Methylphenidate (MPH), a central nervous system stimulant, is the most commonly prescribed medication for children suffering from ADHD. However, not every individual with ADHD responds well to every medication [49]. In 2007, the European Commission formed the ADDUCE (Attention Deficit Hyperactivity Disorder Drugs Use Chronic Effects) project. In its summary, ADDUCE mentions the most common side effects of MDP, which include sleeplessness, nervousness, reduced appetite, headache, abdominal pain, tachycardia, and changes in blood pressure and heart rate. Rarer effects include reduced weight gain and growth reduction occurring with prolonged use [50]. Additional side effects of ADHD medications include sleep and eating problems, tics, mood changes, and rebound headaches. Some of these side effects (e.g., sleepiness, nervousness, growth reduction) are closely resembling SRBD symptoms. Treating children suffering from SRBD with ADHD medications may not only be unnecessary but can cause an aggravation of their disorder [42,51].

Although a recommendation to screen children with hyperactive behavior, inattentiveness, disruptive behavior, or learning disabilities for SRBD and other sleep disorders exists [10,47,52,53], many children remain misdiagnosed. In the present study, about 17% of the SRBD-positive children were using medications for ADHD. None of these children has been previously diagnosed with SRBD. Such a high percentage is alarming. Plausibly, some of the children might have been suffering from SRBD rather than from ADHD and should have been treated in a different way. To avoid misdiagnosis and/or maltreatment, it is important that child caregivers carefully evaluate each case to determine whether the child’s symptoms are part of ADHD (and/or side effects of prescribed ADHD medications) or are possible signs of SRBD.

Four factors emerged as possible predictors of SRBD in children at the age 4–12: taking medications for ADHD increased the odds of SRBD in children by over seven times, non-continuous sleep increased the odds of SRBD by six times, mouth breathing increased the odds by almost five times and snoring increased the odds by over three times. Few simple yes/no questions (does the child snore? Does s/he sleep well at night? Does s/he breath through his/her mouth? Does s/he take medications for ADHD?) can serve as indicators that a child might have an SRBD predisposition. If a suspicion arises, further evaluation is recommended.

The etiological factors of SRBD may be physiological (e.g., BMI), anatomical (i.e., enlarged adenoids, narrow arches, high soft palate), or combined. In order to prevent deleterious impacts on the child’s general health and quality of life, it is crucial to adopt a wide perspective of the problem. An interdisciplinary approach should involve the child’s parents (report of snoring, daytime sleepiness, behavioral problems) and medical caregivers who are in the first line of the child’s health care, such as pediatric physicians, dentists, orthodontists, public health professionals, and school nurses.

The present study was carried out on a limited range of ages, and the number of subjects in two examined groups (non-SRBD and positive SRBD) was imbalanced. SRBD diagnosis was based on questionnaires and not polysomnography, which is the gold standard for SRBD diagnosis. Additionally, the criterion to identify SRBD-positive children was set according to Chervin et al., which defined a cutoff point of ≥8 for children with excessive negative intrathoracic pressures and children with obstructive sleep apnea [21]. However, although the PSQ has been shown to have a test sensitivity of 0.85 and a specificity of 0.87 for identifying those with moderate and severe sleep-disordered breathing [26], a recent publication showed that children with mild SRBD may still manifest elevated scores for inattention and hyperactivity [54]. Therefore, further studies with a larger number of SRBD-positive children and more accurate SRBD definitions are recommended to further address this issue.

## 5. Conclusions

Taking into account the health risks associated with untreated SRBDs, child caregivers should develop a higher awareness of the possible presence of the syndrome. Pediatricians, pediatric dentists, school nurses, and other professionals involved with childcare should actively inquire about disturbed sleep, medications for ADHD, snoring, and mouth breathing among their young patients. Initial screening through a few simple questions may result in raising red flags that can assist in the early detection of SRBD in children and lead to proper diagnosis and treatment. When a concern about SRBD arises, the PSQ questionnaire can be used. Such a simple procedure can help caregivers to identify children at high risk of SRBD and refer them to a sleep specialist.

## Figures and Tables

**Table 1 jcm-11-05570-t001:** Study groups.

Group	Non-SRBD *	SRBD Positive	Test	*p* **
No. (%)	203 (89.4%)	24 (10.6%)		
% female	50%	52.2%	Chi-square	NS
Age	6.8 ± 2.4	6.9 ± 2.9	*t*-test	NS
BMI *	15.85 ± 2.69	17.41 ± 4.72	*t*-test	NS
Number of children in the family	3.7 ± 2.0	3.6 ± 3.0	*t*-test	NS
Week of birth	39.04 ± 1.67	38.78 ± 1.66	*t*-test	NS
Weight at birth	3.22 ± 0.52	3.06 ± 0.72	*t*-test	NS
**PSQ * score**	**2.04 ± 2.05**	**10.20 ± 1.81**	***t*-test**	**<0.001**

* Variables as follows: SRBD—sleep-related disordered breathing, BMI—body mass index, PSQ—Pediatric Sleep Questionnaire; ** significant differences marked in bold, NS—non-significant.

**Table 2 jcm-11-05570-t002:** Children general health status.

	Group	Non-SRBD * (Percentage)	SRBD Positive (Percentage)
Variable			
Background disease		13.8%	33.3%
Asthma		9.4%	16.7%
Hearing problems		3.9%	16.7%
Enlarged tonsils		3.4%	20.8%
Atopic dermatitis		2%	4.2%
Other non-common diseases		5.9%	12.5%
Hospitalization in the past		14.3%	25%
Hospitalization cause: - For surgery Other cause		6.4% 7.9%	16.7% 8.3%
Surgery in the past		6.4%	20.8%
Surgery type - Myringotomy - Tonsillectomy - Myringotomy and tonsillectomy Other surgery		1.5% 2.0% 1.5% 1.5%	8.3% 4.2% 8.3% 0%
Medications in general		5.9%	25.0%
Medication type: - For ADHD * - For asthma Other medications		2.5% 3.0% 1.0%	16.7% 4.2% 12.5%
Known allergies		3.4%	16.7%
Respiratory allergy		3%	12.5%
Reflux		2%	0%
**Non-continuous sleep ****		**15.3%**	**45.8%**
Bruxism		12.3%	12.5%
**Developmental delay ****		**5.9%**	**29.2%**
Behavioral disorders		7.4%	29.2%

* Variables as follows: SRBD—sleep-related breathing disorder, ADHD—attention deficit hyperactivity disorder; ** significant differences marked in bold (chi-square, *p* < 0.001).

**Table 3 jcm-11-05570-t003:** Children oral behaviors and habits.

	Group	Non-SRBD * (Percentage)	SRBD Positive (Percentage)
Variable			
Drinking at night		11.3%	16.7%
Finger sucking		3.0%	8.3%
Use of pacifier		6.4%	4.2%
Nail biting		10.3%	12.5%
**Mouth breathing ****		**11.8%**	**54.2%**
Teeth grinding		10.3%	12.5%
**Snoring ****		**13.3%**	**41.7%**

* Variables as follows: SRDB—sleep-disordered breathing; ** significant differences marked in bold (chi-square, *p* < 0.001).

**Table 4 jcm-11-05570-t004:** Parental self-report.

Parental Self-Report	Parents of Non-SRBD * Children (Percentage)	Parents of Children with SRBD (Percentage)	*p* **
Parental possible bruxism	20.2%	41.7%	NS
**Parental snoring**	**52.2%**	**83.3%**	**<0.005**
Parental respiratory allergy	23.2%	37.5%	NS
Parental SRBD	15.3%	8.3%	NS

* Variables as follows: SRBD—sleep-related breathing disorder; ** significant differences marked in bold (chi-square, *p* < 0.01), NS—non-significant.

**Table 5 jcm-11-05570-t005:** Logistic regression.

Variable	B	S.E.	Wald	df	Sig.	ODDs Ratio	95% C.I.	
							Lower	Upper
Non-Continuous sleep	1.79	0.52	11.61	1	0.001	6.046	2.148	17.020
Taking medications for ADHD *	1.99	0.88	5.13	1	0.023	7.357	1.309	41.332
Mouth breathing	1.54	0.54	8.04	1	0.005	4.669	1.610	13.541
Snoring	1.15	0.56	4.12	1	0.042	3.174	1.042	9.668

* ADHD—attention deficit hyperactivity disorder.

## Data Availability

The datasets generated and/or analyzed during the current study are available from the corresponding author upon reasonable request.

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
