# Peer review of "Sleep-Related Breathing Disorders in Children—Red Flags in Pediatric Care"

_jcm, 2022, doi:10.3390/jcm11195570_

Round 1

Reviewer 1 Report

I have reviewed the article "Sleep related breathing disorders in children-red flags in pediatric care".

The subject is very interesting and still has many concerns that need to be resolved. The present work, although it does not provide great new news, is a good work carried out with a correct methodology and correctly written.

However, some small corrections should be made:

Material and Methods:

Line 105:

The exclusion criteria: “Exclusion criteria were children younger or older than the defined age span” should be removed as it is perfectly defined by the age inclusion criterion.

Results.

 In tables 2 and 5 replace “continuous sleep” by “non-continuous sleep”.

 Discussion

-       Line 221. The sentence "Clinical manifestations and polysomnographic findings in children with SRBD differ from those in adults with SRBD [38]", does not provide any relevant information. It should be deleted or the information of the difference between adults and children should be expanded.

Author Response

Reviewer No.1: Point by point answer

 Dear Reviewer, we would like to thank you for your time and effort in providing a thorough review and for all your valuable comments.

Follows our point-by-point answers to your suggestions:

Material and Methods:

Line 105:

The exclusion criteria: “Exclusion criteria were children younger or older than the defined age span” should be removed as it is perfectly defined by the age inclusion criterion.

Response: The sentence was removed as requested from the exclusion criteria(now in line 107 due to other corrections)

Results.

In tables 2 and 5 replace “continuous sleep” by “non-continuous sleep”.

Response: “continuous sleep” was replaced by “non-continuous sleep” in table 2 with a change in % accordingly and in table 5.

Discussion

-       Line 221. The sentence "Clinical manifestations and polysomnographic findings in children with SRBD differ from those in adults with SRBD [38]", does not provide any relevant information. It should be deleted or the information of the difference between adults and children should be expanded.

Response: The sentence was deleted, and the reference list was corrected accordingly.

Reviewer 2 Report

This is correctly presented study, based on simple clinical methods that should be more frequently used by the clinicians. Some editorial remarks:v. 120 “…teeth or snores during sleep” – should be rather  “…teeth during sleep or snores”, v. 259, 260 – italics here should be omitted. List of bibliography should be prepared according to the rules.

Author Response

Reviewer No.2: point-by-point answers

Dear reviewer,

We would like to thank you for your time and your thorough report.

Follows is our point-by-point answers

 Some editorial remarks:

1)v. 120 “…teeth or snores during sleep” – should be rather “…teeth during sleep or snores”

Response: The sentence was corrected as suggested please consult lines 122-123.

2)v. 259, 260 – italics here should be omitted. Response: text was checked for unnecessary Italics.

3) List of bibliography should be prepared according to the rules. Response: the whole Reference list was edited according to the special issue rules.

Reviewer No.2: point-by-point answers

Dear reviewer,

We would like to thank you for your time and your thorough report.

Follows is our point-by-point answers

 Some editorial remarks:

1)v. 120 “…teeth or snores during sleep” – should be rather “…teeth during sleep or snores”

Response: The sentence was corrected as suggested please consult lines 122-123.

2)v. 259, 260 – italics here should be omitted. Response: text was checked for unnecessary Italics.

3) List of bibliography should be prepared according to the rules. Response: the whole Reference list was edited according to the special issue rules.

Reviewer 3 Report

Congratulations, interesting study. I find it very well written.

Regarding the main question addressed by the paper: after reviewing the literature in the field I find it relevant due to that fact that it reunites several factors which I could not find in other articles.

Most of the already published articles are dealing with separate etiology, which is why it has the factor of novelty.

The paper is clear and easy to read, and the statistics is appropriate.

The conclusions address the aims which are cited in the beginning of the article, and give a simple way of recognising possible problems and if necessary continuing with other clinical investigations if needed.

In terms of references maybe some minor corrections can be done, in terms of editing. 

Author Response

Reviewer No.3:

Dear reviewer,

We would like to thank you for your time, effort, and kind words.

The whole reference list was edited and checked according to the special issue rules.
